# CRISPR-Cas, a Revolution in the Treatment and Study of ESKAPE Infections: Pre-Clinical Studies

**DOI:** 10.3390/antibiotics10070756

**Published:** 2021-06-22

**Authors:** Manuel González de Aledo, Mónica González-Bardanca, Lucía Blasco, Olga Pacios, Inés Bleriot, Laura Fernández-García, Melisa Fernández-Quejo, María López, Germán Bou, María Tomás

**Affiliations:** 1Microbiology Department—A Coruña University Hospital (CHUAC), 15006 A Coruña, Spain; manugo04@gmail.com (M.G.d.A.); monica.gonzalez.bardanca@sergas.es (M.G.-B.); Melisa.Fernandez-Quejo.Mateos@sergas.es (M.F.-Q.); german.bou.arevalo@sergas.es (G.B.); 2A Coruña Biomedical Research Institute (INIBIC), University of A Coruña (UDC), 15006 A Coruña, Spain; luciablasco@gmail.com (L.B.); olgapacios776@gmail.com (O.P.); bleriot.ines@gmail.com (I.B.); laugemis@gmail.com (L.F.-G.); maria.lopez.diaz@sergas.es (M.L.); 3Study Group on Mechanisms of Action and Resistance to Antimicrobials (GEMARA) on Behalf of the Spanish Society of Infectious Diseases and Clinical Microbiology (SEIMC), 28003 Madrid, Spain; 4Spanish Network for Research in Infectious Diseases (REIPI), 41071 Seville, Spain

**Keywords:** CRISPR-Cas, ESKAPE pathogens, treatment

## Abstract

One of the biggest threats we face globally is the emergence of antimicrobial-resistant (AMR) bacteria, which runs in parallel with the lack in the development of new antimicrobials. Among these AMR bacteria pathogens belonging to the ESKAPE group can be highlighted (*Enterococcus* spp., *Staphylococcus aureus*, *Klebsiella pneumoniae*, *Acinetobacter baumannii*, *Pseudomonas aeruginosa* and *Enterobacter* spp.) due to their profile of drug resistance and virulence. Therefore, innovative lines of treatment must be developed for these bacteria. In this review, we summarize the different strategies for the treatment and study of molecular mechanisms of AMR in the ESKAPE pathogens based on the clustered regularly interspaced short palindromic repeats (CRISPR) and CRISPR-associated (Cas) proteins’ technologies: loss of plasmid or cellular viability, random mutation or gene deletion as well directed mutations that lead to a gene’s loss of function.

## 1. CRISPR-Cas: An Adaptive Prokaryotic Immune System

Clustered regularly interspaced palindromic repeats (CRISPR) and the subsequent CRISPR-associated (Cas) proteins constitute an adaptive immune system in both bacteria and archaea. They were first identified in *Escherichia coli* in 1897 by Ishino and colleagues [1] and renamed as CRISPR by Jansen and colleagues [2]. However, it was Spanish microbiologist Francis Mojica who deepened their function and significance while studying the halophilic archaeon *Haloferax mediterranei* [3,4,5]. Finally, Doudna and Charpentier’s groups unraveled the process through which CRISPR-derived RNAs (crRNA) are processed, directed by the transactivating CRISPR RNAs (tracrRNA) [6,7,8,9]. Both of them were recently awarded with the Nobel Prize in Chemistry 2020 “for the development of a method for genome editing”.

The structure of a CRISPR-Cas system consists of an array of exogenous DNA sequences (known as spacers) flanked by similarly sized and identical direct and inverted repeats (known as palindromic repeats), as well as upstream *cas* genes. These spacers derive from foreign phages and/or plasmids entering the bacterial cell, and its chronologic insertion into the CRISPR array implies the acquisition of a “memory fragment” of those invaders. If the exogenous DNA enters the cell in the future, the bacterium specifically recognizes the DNA sequences matching its spacers and cleaves them through the Cas proteins’ nuclease activity, serving as a sequence-specific bacterial defense [10].

To avoid autoimmunity caused by the bacterium targeting its own DNA, protospacer adjacent motifs (PAMs) are 2–6 base pair (bp) sequences located within the invader DNA, near the Cas protein’s target. Without a PAM sequence, the CRISPR-Cas system is unable to bind to the target sequence and induce the strand separation required for the nuclease domain to act [9,11,12].

CRISPR-Cas systems can be classified into two main classes, each one divided into different types. Class I encompasses types I, III and IV, in which the identification of the target sequence and its cleavage is handled by a set of different proteins (Cas5, Cas7, SS, etc.). For class II, both identification and cleavage of the target sequence is performed by a single enzyme: Cas9 for type II, Cas12 for type V and Cas13 for type VI (Figure 1).

CRISPR-Cas system’s activity is usually classified into three different steps, as reviewed by Strich and Chertow in 2018: adaptation, crRNA maturation and interference [11]. Firstly, PAM sequences within the exogenous DNA are recognized and cleaved into small fragments by Cas1, Cas2 and Cas4 enzymes. This process, known as adaptation, consists of the sequential integration of the resulting DNA fragments into the CRISPR array, next to the AT-rich motif. This requires the intervention of further enzymes, such as the integrated host factor (IHF), Cas9, Csn2 or tracrRNA, and their participation depends on the CRISPR-Cas system’s class. Then, crRNA maturation occurs when the CRISPR array is transcribed, and the resulting pre-crRNA is processed into individual crRNAs by the Cas proteins, which specifically recognize repeats. The secondary structure of the transcript given by its palindromic repeats, which form a loop, is essential for spacer identification and cleavage by Cas proteins in systems I-E, V-A and VI-A. For type III-A, a Cas6 dimer is responsible for crRNA maturation, and no loop formation is needed, whereas in type type II-A, RNase III cleaves the CRISPR transcript upon formation of a complex between the spacer, a tracrRNA and a Cas9 protein. Finally, interference takes place when the crRNA:Cas protein complex (crRNA:tracrRNA:Cas protein complex in type II-A) is directed to the invading DNA, specifically recognizes the target sequence and cleaves it, causing a fatal double-strand break [11].

## 2. CRISPR-Cas: A New Concept of Antimicrobials

During the last decades, the slow rate of new antimicrobial development when compared to the rapidity through which microorganisms (including parasites, virus and, specially, fungi and bacteria) acquire resistance to them has been a hot topic in microbiology. Costs derived from drug research and development, together with the limited duration of the antimicrobial treatment until the resolution of the infection, dramatically reduce the benefits of this industry. In addition, the risk of an eventual loss of efficacy due to the emergence of AMR makes the antimicrobial drug industry unappealing to pharmaceutical companies, which tend to focus on more profitable topics such as chronic diseases [10,14].

In this context, new approaches against pathogenic bacteria have emerged, with different mechanisms of action: antimicrobial peptides, metal nanoparticles, bacteriophages and gene edition tools [15]. The latter are of great interest due to their ability to target and cleave precise sequences within the bacterial genome in a species-specific manner, resulting in antimicrobials with the narrowest possible spectrum. These gene edition tools are zinc fingers [16,17], transcription activation-like effector nucleases (TALENs) [18], peptide nucleic acids [19], RNA interference (RNAi) [20] and CRISPR-Cas systems [10,21]. In the first three, specificity is given by protein–DNA interactions, which require protein engineering for its development. This makes it challenging, expensive and time-consuming to reshape the effector proteins in order to adapt them to new targets. However, CRISPR-Cas specificity is achieved through RNA-DNA interactions, with RNA engineering being much more affordable and thus a perfect candidate for a new concept of antibiotics based on gene edition [11].

CRISPR-Cas can be used following three general strategies: (i) it can be directed to cleave species-specific genes to treat acute infections, resulting in a deployment of the bacteria of interest while maintaining the host’s microbiome unaltered [22]; (ii) it can be directed to cleave drug-resistance genes, eliminating bacteria harboring them while maintaining the viability of the wild-type susceptible clones and thus decolonizing patients [23]; or (iii) it can be directed to modify or silence resistance genes, introducing mutations that cause resistance genes’ loss of function while maintaining bacterial viability in a process known as resensitization [24] (Figure 2).

Within all Cas proteins, the ones which were used the most to address AMR were the following: (a) Cas9, which specifically recognizes its target and cleaves it, causing a double-strand break [25]; (b) dCas9, a defective Cas9 protein lacking the double-strand nuclease activity which specifically recognizes its target and stays attached to that region, hampering the binding of the RNA polymerase and thus the formation of the transcription preinitiation complex [26]; (c) nSpCas9:rAPOBEC1, a Cas9 protein without nuclease activity fused to a deaminase, which causes the conversion of cytidine bases into thymine ones, thus creating a stop codon [24]; and (d) Cas13a, an RNA-specific endonuclease which indiscriminately cleaves RNA fragments upon activation by the recognition of its specific DNA sequence [27] (Figure 3).

Among all CRISPR-Cas types, the most broadly used for genetic engineering is type II due to its simplicity and the fact that it employs a single nuclease with two catalytic domains for interference (Cas9), in contrast to the protein complex required for the types among class 1 [11,28]. The general scheme through which type II CRISPR-Cas system can be used to target bacterial genes is as follows: firstly, a chimeric sgRNA (an artificial RNA construct which fuses crRNA and tracrRNA) is designed to be complementary to the target sequence in the bacterial gene of interest. The chosen sequence has to be unique within the bacterial genome and mobilome so that it only attaches to the desired gene, as well as possessing a PAM. Then, sgRNA:Cas9 complexes are recruited and cleave the target sequence within the bacterial genome, producing blunt ends [29]. This chromosomal double break poses a serious risk for the bacterial cell’s integrity, and it can be mended through nonhomologous end-joining repair (NHEJ) [30], which is error prone and often leads to loss of protein function by inserting aleatory nucleotides or even the loss of the cell’s viability [10]. The cytotoxicity of targeting chromosomal self-genes has been previously shown by Vercoe and colleagues in *Pectobacterium artrosepticum* strains [31]. In this study, the bacteria’s own CRISPR machinery was exploited by introducing expression vectors which coded for specific crRNAs targeting three nonessential chromosomal genes. As a result, a reduction of a hundred-thousand-fold in viable bacterial counts was observed, as well as the filamentation of surviving cells. In another study, Hullahali and colleagues demonstrated that the growth of *Enterococcus faecalis* strains is impaired when its own CRISPR machinery is modified to target self-genes [32]. However, this cytotoxicity can be prevented by inserting into the cell an artificial DNA fragment with a copy of the target gene, serving as template for homology-directed repair (HDR) [33] instead of NHEJ. If this copy is engineered with desired mutations, the targeted bacteria would be able to acquire them through a recombination process, resulting in a knockdown of the gene of interest while maintaining the cell’s viability (Figure 4).

## 3. CRISPR-Cas: A Species-Specific Treatment for ESKAPE Infections

Morbidity and mortality associated with bacterial resistance is restlessly increasing. The European Society for Clinical Microbiology and Infectious Diseases (ESCMID) has delimited three different categories to classify resistant bacteria: multidrug-resistant (MDR) bacteria, when there is in vitro resistance to at least one agent in three or more antimicrobial categories; extensively drug-resistant (XDR) bacteria, when there is resistance to at least one agent in all but two or fewer categories; and pandrug-resistant (PDR) bacteria, where there is resistance to all antimicrobials [34].

According to the Centers for Disease Control and Prevention (CDC), in the year 2019, more than 2.8 million infections occurred in the US with an antibiotic-resistant causative agent, of which 35,000 resulted in the patient’s death [35]. Additionally, the estimated cost to treat MDR infections in the US was calculated to be more than 4.6 billion dollars in 2017 [36]. Regarding Europe, MDR infections are responsible for about 33,000 deaths annually, with an estimated cost of 1.1 billion euros (1.3 billion dollars) [37].

In this context, in 2008, Rice set a list of six main pathogens for which the development of new antibiotics was (and still is) crucial: the ESKAPE bacteria. These are *Enterococcus faecium*, *Staphylococcus aureus*, *Klebsiella pneumoniae*, *Acinetobacter baumannii*, *Pseudomonas aeruginosa* and *Enterobacter cloacae* [38]. Organizations such as the Infectious Disease Society of America (IDSA) have emphasized the need of joining efforts to tackle these bacteria due to their virulence, prevalence in nosocomial environments and drug resistance [39,40].

In this review, we focus on research directed to address AMR in the ESKAPE group by the edition of drug resistance or virulence genes with the different CRISPR-Cas technology available. Moreover, we also review the literature focused on the use of the CRISPR-Cas technology to study the molecular mechanisms of AMR (Table 1). For those purposes, a profound search in the Pubmed database was performed between 15 December 2020 and 31 March 2021. The included words were: CRISPR, treatment, ESKAPE, *Enterococcus* spp., *Staphylococcus aureus*, *Klebsiella pneumoniae*, *Acinetobacter baumannii*, *Pseudomonas aeruginosa* and *Enterobacter* spp.

### 3.1. Enterococcus faecium and E. faecalis

*Enterococcus* spp. is a genus of Gram-positive cocci arranged in pairs or short chains. Despite being part of the gastrointestinal microbiota, two species are often found to cause infection: *E. faecalis* and *E. faecium* [41]. Their ability to survive on inert surfaces for long periods of time has made them an important issue in hospital-acquired infections, and mutations and/or overproduction of a penicillin binding protein (PBP) of the class B, known as PBP5, confer them intrinsic resistance to the majority of β-lactams. The exception is ampicillin, which is effective in the majority of *E. faecalis* strains [42]. Enterococci are also intrinsically resistant to aminoglucosides, thus reducing the possible therapeutic options. In addition to that, in the last years the continuous increase in vancomycin-resistant enterococci (VRE) has been of special concern, with the number of infections in Europe almost doubling from 2007 to 2015, according to the ECDC annual report [43].

To date, we only have knowledge of a single gene edition study in *E. faecium* using CRISPR technology. In 2020, de Maat and colleagues harnessed the high recombination rates in *E. faecium* to insert two copies of the green fluorescent protein (GFP) into the macrolide resistance gene *msrC* [44]. This was performed by firstly transform the vancomycin-resistant *E. faecium* E745 clinical strain with a pVLP3004 plasmid encoding the Cas9 protein and a tracrRNA. Afterward, a second plasmid, named pVDM1001, was used, encoding the specific crRNA and a donor DNA to serve as a template for HDR. This dual plasmid strategy is an adaptation from Oh and Van Pijkeren’s work with *Lactobacillus reuteri* [45]. Successful edition was assessed by fluorescence measuring after plasmid curing; however, researchers did not study macrolide MICs after GFP insertion; therefore, unfortunately, resensitization could not be tested.

Furthermore, several studies have been published regarding another enterococcal species: *E. faecalis*. An interesting approach to treat enterococcal infections consists of harnessing the bacterium’s own CRISPR machinery. This was studied by Dr. Palmer’s group, who observed that Type II CRISPR2 orphan locus, which lacks the *cas* genes, could be reactivated to target pheromone-responsive plasmids (PRP) in the presence of Cas9 enzymes [46]. PRPs are *E. faecalis* specific plasmids in which high-frequency conjugation is enhanced by the constant production of small signaling peptides by recipient cells and often harbor resistance and virulence factor genes [47]. In their study, Price and colleagues activated the CRISPR2 locus, which lacks the *cas* genes, against the PRP pCF10 by introducing into the MDR *E. faecalis* T11 strain a CRISPR1-derived Cas9 enzyme [46]. In addition to that, they analyzed the effect of both restriction-modification and CRISPR-Cas3 systems against the pAM714 PRP. A year later, this group reactivated the same orphan type II CRISPR2 system in the vancomycin-resistant *E. faecalis* V583 strain to target mobile genetic elements. The authors noted that, despite what was previously published, in *E. faecalis*, a spacer and its corresponding target could temporarily coexist. In the presence of selective pressure (exposure to 15 µg/mL chloramfenicol and 50 µg/mL erythromycin), the toxic type II CRISPR2 spacers were gradually lost, whereas in the absence of antimicrobials, it was the CRISPR targets that were eliminated [32].

Finally, in 2019, Rodrigues and colleagues integrated a CRISPR system targeting *tetM* and *ermB* genes—which confer resistance to tetracycline and erythromycin, respectively—into a PRP, named as pKH88[sp-tetM] and pKH88[sp-ermB] [48]. These plasmids were transmitted by conjugation using *E. faecalis* CK135 as a donor strain and *E. faecalis* OG1SSp as a recipient strain, successfully removing antibiotic resistance in vitro. Afterward, using an in vivo C57BL6/J mouse model, it was seen that despite the low recombination rate obtained, transconjugants who successfully acquired the PRP became unable to gain erythromycin-resistance genes. This led the authors to suggest the possibility of using probiotic *E. faecalis* strains harboring these PRPs to hamper patient’s colonization by resistant *E. faecalis* strains.

### 3.2. Staphylococcus aureus

*Staphylococcus aureus* is a species of Gram-positive, coagulase and catalase positive cocci arranged in clusters and one of the main bacterial pathogens. In the last decades, the spread of methicillin-resistant *S. aureus* (MRSA), which harbors the *mecA* gene located in the SCCmec (staphylococcal cassette chromosome), has raised special concern [49]. The *mecA* gene is responsible for the production of PBP2a, which has low affinity for β-lactam antibiotics and results in resistance to penicillins, cephalosporins—except for the novel antibiotics ceftaroline and ceftobiprole—and carbapenems [50].

In 2014, Bikard and colleagues studied the ability of CRISPR-Cas systems to eliminate specific strains of *S. aureus* from mixed cultures without affecting other strains of the same species through phagemids [23]. Phagemids are genetic-engineering constructs which fuse a plasmid’s replication origin with filamentous phage’s coat proteins and related genes. For those purposes, a phagemid (pDB121) was designed to target the *mecA* gene in a 50/50 mix with a clinical MRSA strain (USA300φ) and a RN4220 strain lysogenized—and thus phage-immunized—with φNM1, named RNφ. The MRSA proportion dramatically decreased from the original 50% to a 0.4%. However, the authors remarked two main disadvantages to this study: the inability of phagemids to produce copies of themselves, making it necessary to inoculate a greater number of phagemids than the number of target cells, and its unknown display in a more complex environment, such as diverse microbial populations within living organisms [23].

In 2017, Guan and colleagues transformed *S. aureus* AH1 strains with engineered plasmids (pLI-158 and pLI-252), encoding a CRISPR-Cas system with spacers targeting the *mecA* gene [51]. Upon transformation, 95% of the bacterial population was eliminated, thus proving the cytotoxicity caused by targeting the host’s own chromosome. Interestingly, the surviving 5% of transformants had overcome CRISPR’s fitness cost through three different ways: target deletion (87.5%), mutations in the cas genes (10.2%) and spacer deletion (2.3%). As expected, those strains in which the *mecA* gene was eliminated were resensitized to oxacillin, whereas the other two remained resistant.

On the same year, Liu and colleagues inserted the *ermR* gene in the *S. aureus* RN4220 strain into the *mecA* cassette [52]. Researchers engineered a plasmid, pLQ-KI-ermR, encoding an *ermR* targeting sgRNA, the *cas9* gene and a donor DNA, which served as a template for HDR. The achieved gene-edition efficiency was 43–96%, claimed by the authors to be superior to the allelic replacement’s efficiency (17–80%). Finally, plasmid curation was achieved by introducing a temperature-sensitive replicon into the plasmid and afterward incubating transformants at 42 °C, reaching an efficiency of 99.5%. Despite the successful gene edition, researchers did not study MICs to erythromycin or oxacillin of the edited strains.

Simultaneously, Park and colleagues removed any virulence factors from the temperate phage φSaBov and engineered it to carry a CRISPR-Cas system targeting the *nuc* gene, an *S. aureus* species-specific gene which encodes a thermostable nuclease [25]. The resulting phage was produced in RF122 *S. aureus* strains and termed φSaBov-Cas9-nuc. In their in vitro studies, the authors achieved an almost total decolonization from CTH96 *S. aureus* strains, recovering no viable cells after treatment with 100 multiplicities of infection (MOI, the ratio between the infecting particles and the host cells). By contrast, the same phage lacking the CRISPR-Cas system had an insignificant effect, which was attributed by the authors to the phage’s own lytic cycle. In their in vivo studies, the authors used a skin infection model in C57BL/6 mice and found more than two orders of magnitude in CFU reduction when applying φSaBov-Cas9-nuc embedded into a hydrogel. This was not observed when the phage was applied directly into the dry skin, due to the latency of bacterial metabolic activity in inert surfaces, which is needed for the CRISPR-Cas system’s functionality. Finally, in order to increase the host’s specificity, φSaBov-Cas9-nuc’s tail fiber protein was complemented with that from the broader-spectrum phage φ11, resulting in a specificity extended to the human pathogenic clones ST1, ST5, ST8 and ST36 [25].

In 2019, the same group studied the effects of their engineered phage φSaBov-Cas9-nuc on biofilms, both in vitro and in vivo [53]. For tracking purposes, biofilm forming ATCC 6538 *S. aureus* strain was modified by homologous recombination to incorporate a green fluorescent protein (GFP) into its chromosome. In their experiments, Cobb and colleagues compared the in vitro efficacy of vancomycin, fosfomycin and the CRISPR-Cas system harbored in the phage φSaBov-Cas9-nuc. Whereas vancomycin showed no effect on biofilm even at high concentrations (1024 µg/mL), fosfomycin (64 µg/mL) and φSaBov-Cas9-nuc (1 × 10^8^ pfu/mL) successfully cleared it. Furthermore, the phage therapy was shown to be superior to fosfomycin by fluorescent tracking of the GFP. A rat osteomyelitis model was afterward developed by inoculating the ATCC 6538-GFP *S. aureus* strain in a screw which was then placed into the rat’s femur. After 7 days of the procedure, the infection site was treated with an alginate gel containing 3 g of fosfomycin, 3 × 10^7^ pfu/mL of phage or both fosfomycin and phage. Whereas in the surrounding soft tissue all three treatments were shown to yield lower bacterial counts than the control (alginate alone), only fosfomycin was effective for the femur. This is believed to be due to the lower phage concentration achieved in the site of infection, unable to reach the quantity of 1 × 10^8^ pfu/mL previously described to be effective. This was due to the small volume of alginate that could fit into the bone’s incision and the higher density of the phage-containing gel, resulting in lower phage concentrations [53].

In 2019, Wu and colleagues resensitized *S. aureus* ATCC 6538 strains to lysostaphin [26], a multiple-catalytic activity enzyme which specifically cleaves *S. aureus* cell wall’s interpeptide crossbridges [54]. Resistance to this bacteriolytic enzyme is developed by modifying the negative charge of the cell wall teichoic acids, thus hampering lysostaphin adhesion to the bacterial surface. In this study, researchers constructed a plasmid encoding a nuclease-defective Cas9 protein (dCas9) from *S. pyogenes* and several sgRNAs targeting the *tarO*, *tarG* and *tarH* genes, responsible for the teichoic acids’ synthesis. These genes are regulated in a cascade process, ultimately controlled by the *tarO* gene [54]. Approximately, a 38% reduction in transcription of these genes was observed by RT-PCR, resulting in ≈1 log kill of cells when incubated with 3 µg/mL lysostaphin and in an extent comparable to the wild-type susceptible strains [26]. However, *tarO* deletion was shown to cause a greater increase in *S. aureus* lysostaphin susceptibility, suggesting that dCas9 gene silencing was not complete. To solve this issue, the authors propose to enhance the repressing efficiency by increasing the dCas9 copy number in the transforming plasmid, targeting different DNA strands depending on the promoter’s position and constructing chimeric dCas9-transcriptional repressors. This strategy of employing a dCas9 enzyme has been also studied by Chen and colleagues to edit several genes in *S. aureus* not related to drug resistance (*cmyR*, *agrA*, *cntA*, *murR*, *agrA* and *sasE*) [55].

Moreover, in 2017, Kang and colleagues for the first time studied Cas9 proteins covalently linked to branched polyethyleneimine (bPEI), a cationic polymer, to enhance CRISPR-Cas uptake by the target cells [56]. This nonviral delivery method was proven to be more effective at targeting the *mecA* gene in *S. aureus* CCARM 3798, 3803 and 3877 strains than the CRISPR-Cas system alone or carried by lipofectamine, an artificial transfection reagent used for mammalian cells. Furthermore, the amount of bPEI required to pack the CRISPR-Cas constructs (SpCas9-bPEI) was considerably lower than that for lipid formulations, claimed by the authors to be crucial to reduce toxicity and allow the delivery of higher concentrations.

Furthermore, in 2020, Kiga and colleagues studied the effect of a different kind of Cas enzyme, Cas13a, packaged in a bacteriophage capsid [27]. This protein’s peculiarity resides in its ability to indiscriminately cleave single-stranded RNAs whenever the enzyme recognizes a viral transcript, thus limiting the phage’s replication by stopping the bacterial metabolism and growth. Hence, the activation of the Cas13a protein can be lethal for the targeted bacteria. For this experiment, a CRISPR-Cas13a construct targeting the *mecA* gene (pKLC-SP_mecA) was inserted into the *S. aureus* 80α phage’s capsid. This was studied in different *S. aureus* strains with resistance (MRSA USA300 strain) or susceptibility to oxacillin (USA300-∆*mecA* and RN4220 strains), finding out that only the oxacillin-resistant had a significantly impaired growth.

Finally, regarding the study of AMR, in 2018, Penewit and colleagues used the CRISPR-Cas technology not to edit bacterial genes but to eliminate the unedited ones in a process known as CRISPR counterselection, leading to high-throughput recombination rates [57]. A plasmid was introduced into *S. aureus* ATCC 29213 strains encoding a recombinase from *E. faecalis* (EF2132), and cells were transformed with a recombineering oligonucleotide containing a copy of the *rpoB* gene with a single-nucleotide mutation (H481Y), which confers rifampin resistance and a silent mutation that disrupts a PAM motif present in the wild-type gene. Afterward, a plasmid encoding Cas9 proteins and sgRNAs targeting the wild-type *rpoB* gene was introduced, causing double-strand breaks in those cells in which recombination had not occurred. However, in those with successful recombination, the PAM motif was eliminated, and *rpoB* mutation was acquired, meaning that cells gained both CRISPR-Cas immunity and rifampin resistance. Thanks to the temperature-sensitive replicon in which plasmids were engineered, cells could be cured from those external elements after overnight incubation at 43 °C, and selection markers could be lost [57].

### 3.3. Klebsiella pneumoniae

*Klebsiella pneumoniae* is a facultative anaerobe, capsulated, Gram-negative rod belonging to the *Enterobacterales* order and commonly found in the human gastrointestinal tract [58]. The wild-type strains only manifest intrinsic resistance to aminopenicillins (ampicillin, amoxicillin) and carboxipenicillins (ticarcillin and piperacillin) due to the chromosomal β-Lactamase SHV-1 [59]. However, the ability of *K. pneumoniae* to acquire resistance to virtually all approved antimicrobials through a combination of plasmid-borne carbapenemases and other resistance mechanisms is of special concern [60,61,62,63].

In 2018, Wang and colleagues developed two different genetic engineering methods based on CRISPR technology to reverse antibiotic resistance in *K. pneumoniae* [24]. Firstly, they designed a plasmid (pCasKP) harboring the *S. pyogenes cas9* gene and a sgRNA targeting the *fosA* gene, which encodes a glutathione transferase responsible for fosfomycin resistance [64]. Due to the lack of a NHEJ pathway in *K. pneumoniae* and the low yield of HDR observed when a donor DNA was offered as template, another plasmid (pSGKP) with the λ Red recombination system was used. This high-efficiency bacteriophage recombination machinery for double-strand breaks has been studied and well characterized in the last 50 years [65]. Thanks to the combination of both plasmids (pCasKP-pSGKP), the authors resensitized the *K. pneumoniae* clinical strain KP 5573 to fosfomycin with an editing efficiency of 100% (10 out of 10 randomly picked colonies) [24].

Afterward, they designed a second genetic engineering tool (pBECKP) by fusing the Cas9 nickase nSpCas9 with the murine cytidine deaminase rAPOBEC1, creating a chimeric protein capable of recognizing specific sequences within the bacterial genome, inducing single-stranded breaks and converting cytidine bases into thymine ones. This tool was used to create a premature stop codon in the *fosA* gene of *K. pneumoniae* KP5573 strains, leading to the production of a truncated protein and reversing fosfomycin resistance with a 100% efficiency (8 out of 8 randomly picked colonies) [24].

Finally, researchers exploited these two tools to resensitize the hypermucoviscous *K. pneumoniae* clinical strain KPCRE23 to carbapenems by both deleting (through pCasKP-pSGKP) or truncating (pBECKP) the *bla_KPC-2_* carbapenemase gene and the two *bla_SHV_* and *bla_CTX-M-65_* ESBL genes. In the first case, chromosomal *bla_SHV_* deletion yielded an efficiency of 4/12, whereas for plasmid-borne *bla_SHV_* and *bla_CTX-M-65_* genes, no PCR product was observed upon transformation, nor for the wild-type plasmid and nor for the ESBL-deleted plasmid. The authors suggested a plasmid loss due to the critical double-strand break caused by Cas9, opening the possibility for the removal of drug-resistance plasmids by CRISPR-Cas constructs. Lastly, the pBECKP tool was used to generate a stop codon into the *bla_KPC-2_* gene, with an efficiency of 100% (8 out of 8 randomly picked colonies) and a reduction in imipenem MICs from 64 to 1 µg/mL, resulting in KPCRE23 resensitization to carbapenems [24].

Finally, in 2020, Hao and colleagues introduced CRISPR-Cas encoding plasmids (pCasCure) into carbapenem-resistant strains of *K. pneumoniae* to resensitize them to imipenem and meropenem in a very efficient manner. First of all, the KPC-2 encoding plasmid IncFIIK-pKpQIL from *K. pneumoniae* 13001 (ST258) harboring the *bla_KPC-2_* gene was cured with an efficiency ranging from 98.6 ± 2.4% to 100%. Then, the KPC-2 encoding plasmid IcnN from *K. pneumoniae* Kp97_58 (ST111) harboring the gene *bla_KPC-2_* was cured with an efficiency ranging from 96.5 ± 2.4% to 97.9 ± 2.1%. Afterward, the OXA-48-like encoding plasmid p72_X3_OXA181 from *K. pneumoniae* 5193 (ST307) harboring the gene *bla_OXA-48-like_* was cured with an efficiency of 98.6 ± 1.2%. Finally, although the OXA-48 encoding plasmid IncL-pOXA48 from *K. pneumoniae* 49210 (ST23) harboring the gene *bla_OXA-48_* could not be removed due to IS1R-mediated recombination events, the target gene was found to be truncated, and sensitivity to carbapenems was restored. In this last strain, an additional sgRNA targeting the IncL replicon was inserted into the pCasCure CRISPR-plasmid, resulting in a plasmid curing efficiency ranging from 99.3 ± 1.2% to 100 ± 0%. In all of the strains, MICs to carbapenems decreased from >16 mg/L to lower than 0.25 mg/mL, proving a successful resensitization [66].

Finally, regarding the use of CRISPR-Cas to study the molecular mechanisms of AMR, in 2019, Sun and colleagues engineered carbapenem-resistant strains of *K. pneumoniae* to study colistin and tigecycline resistance by using the abovementioned dual-plasmid genome-editing system (pCasKP-pSGKP) [24]. These two antibiotics are considered to be last-resort options in carbapenem-resistant Enterobacterales’ infections. First of all, researchers used the Y4 strain of *K. pneumoniae* (susceptible to colistin, with a MIC of 0.25 mg/L) to target and delete the *mgrB* gene via CRISPR-Cas [67]. This gene is responsible for the production of a transmembrane regulatory protein which downregulates the PhoPQ-PmrAB pathway, a system which modifies de phosphate groups within the lypopolysaccharide’s lipid A by inserting amino-arabinose residues and thus changing the negative charges for positive ones. The elimination of *mgrB* gene results in an accumulation of positive-charged LPS within the bacterial cell wall and thus in colistin resistance due to like charges repulsion [68,69]. Upon *mgrB* deletion through pCasKP-pSGKP, Y4 *K. pneumoniae* strains’ MIC to colistin increased from 0.25 to 16 mg/L, classified as resistant by both antimicrobial susceptibility testing agencies, CLSI (Clinical and Laboratory Standards Institute) [70] and EUCAST (European Committee on Antimicrobial Susceptibility Testing) [71].

On the other hand, the Y17 *K. pneumoniae* strain (resistant to tigecycline, with a MIC of 8 mg/L) was used to analyze the effects of mutations in *tetA* and *ramR* genes. Whereas the *tetA* gene encodes for a tetracycline-specific efflux pump, the *ramR* gene downregulates the AcrAB efflux system via the *ramA* gene [72,73]. In their research work, Sun and colleagues used the pBECKP plasmid to create a stop codon into the *tetA* gene, resulting in a reduction of tigecycline from 8 to 2 mg/L, although it was insufficient to resensitize the strain (EUCAST sets susceptibility below 0.5 mg/L). In addition, *ramA* gene was deleted by using the dual-plasmid pCasKP-pSGKP editing system, increasing tigecycline MICs in *K. pneumoniae* Y17 from 8 to 64 mg/L, thus proving the implications of both genes in tigecycline resistance [67].

### 3.4. Acinetobacter baumannii

*Acinectobacter baumannii* is a strictly aerobic, nonfermentative, Gram-negative cocobacilli of special interest in hospital environments. Their ability to survive on inert surfaces and to adhere to materials such as latex in gloves has made them one of the main etiologic agents in hospital-acquired infections. In addition, a combination of efflux pumps and the impermeability of its outer membrane, together with its ability to acquire mobile genetic elements, confers extremely high drug-resistance rates to these bacteria [74,75].

In *A. baumannii*, little has been studied regarding the use of CRISPR-Cas systems as antimicrobial treatment. In 2018, Karlapudi and colleagues analyzed the *abaI* gene from *A. baumannii* through different bioinformatic tools to design suitable sgRNAs for its silencing [76]. *AbaI* is responsible for the synthesis of acylhomoserine-lactones, well-studied quorum-sensing autoinducers that regulate biofilm synthesis in *A. baumannii* [77,78].

Between 2019 and 2020, Wang and colleagues adapted the previously cited double-plasmid CRISPR tool in *K. pneumoniae* [24] to be used in *A. baumannii* gene editing. Given the intrinsic HDR activity of this species, a single plasmid harboring a CRISPR-Cas system was engineered (pCasAb). However, this recombination activity was found to be insufficient for repairing double-strand breaks per se; therefore, a recombination machinery was engineered in another plasmid as well (pSGAb). This plasmid carried the RecA recombination system, which was found to be more efficient for *A. baumannii* than the λ-Red (previously used in *K. pneumoniae*) and Rac-RecET systems [79]. However, to avoid plasmid loss risk when targeting extrachromosomal genes related to plasmid double-strand breaks, another CRISPR-Cas system was designed. In this case, the Cas9 nuclease was replaced by a chimeric nickase nSpCas9 fused with the murine cytidine deaminase rAPOBEC1 (pBECAb-apr) in an analogous way as it was performed with *K. pneumoniae*. This way, artificial stop codons could be made at desired locations by substituting Cs for Ts, silencing genes without hampering the DNA’s integrity. With this system, three β-lactamase encoding genes in *A. baumannii* XH386 were targeted (*bla_OXA-23_*, *bla_TEM-1D_* and *bla_ADC-25_*), and susceptibility of the resulting mutants to imipenem and sulbactam was tested. Deletion of TEM-1D (labeled as ∆*TEM*) yielded no difference in imipenem susceptibility when compared to the WT, whereas deletion of ADC-25 (labeled as ∆*ADC*) resulted in a 2-fold increase and deletion of OXA-23 (labeled as ∆*OXA*) in an 8-fold increase. When simultaneously targeting more than one β-lactamase, the greatest effect was obtained in the ∆*TEM*∆*ADC*∆*OXA* and ∆*ADC*∆*OXA* mutants, suggesting no significant role of TEM-1D in imipenem’s susceptibility. On the other hand, ∆*ADC*, ∆*OXA* and ∆*ADC*∆*OXA* mutants did not show any reduction in susceptibility of *A. baumannii* to sulbactam, whereas ∆*TEM* mutants displayed a 2-fold increase. The greatest change was observed in the ∆*TEM*∆*ADC*∆*OXA* mutant, with a 4-fold increase in susceptibility to sulbactam. These results showed that TEM-1D was the main β-lactamase responsible for sulbactam resistance, suggesting the CRISPR-based plasmid pBECAb-apr to be a useful tool to elucidate the contribution of the different β-lactamases to drug resistance [80,81].

### 3.5. Pseudomonas aeruginosa

*Pseudomonas aeruginosa* is a strictly aerobic and nonfermenting Gram-negative rod commonly found as an opportunistic human pathogen. As in *A. baumannii*, its high drug-resistance rates, mainly acquired by a combination of low permeability of the outer membrane and active drug expelling out of the cell, are of special concern. The acquisition of mobile genetic elements encoding drug-resistance genes such as β-lactamases also plays an important role in antibiotic resistance. In addition, its ability to colonize patients with chronic diseases (such as cystic fibrosis or bronchiectasis) and form biofilm hampers antibiotic activity and the eradication of the pathogen [82].

In 2019, Xu and colleagues exploited the native CRISPR-Cas system found in the MDR *P. aeruginosa* PA154197 strain, classified as type I-F [83,84]. This strategy has been already used in species in which gene edition is usually inefficient due to poor genetic homeostasis and intrinsic cytotoxicity of heterologous Cas9 proteins, such as clostridia [85,86]. To avoid this issue, it is possible to harness the bacterium’s own CRISPR-Cas machinery by introducing into the cell sgRNA encoding plasmids, whose transcripts direct native Cas proteins to the desired targets.

After confirming the functionality of the native CRISPR-Cas system in the *P. aeruginosa* PA154197 strain, Xu and colleagues designed the plasmid pAY5233, which encoded sgRNAs targeting the *mexB* gene. This gene is responsible for the production of a component of the MexAB-OprM efflux pump in *P. aeruginosa*, which expels molecules such as antimicrobial drugs out of the cell, conferring resistance to them [87]. Targeting the *mexB* gene yielded no transformants, suggesting the potential toxicity of chromosomal double-strand breaks. Afterward, a new strategy was used, targeting this same gene but at the same time introducing into the pAY5233 plasmid, renamed as pAY5235, a donor DNA template for HDR. With this approach, it was possible to achieve deletion of *mexB* and regulatory genes *mexF* and *mexH* with a success rate above 90%, and changes in antimicrobial drug susceptibility were the analyzed. For ∆*mexB* mutants, MICs to antipseudomonal β-lactams were reduced by more than 128-fold (carbencillin), 64-fold (aztreonam, piperacillin/tazobactam), 32-fold (meropenem) and 8-fold (ceftazidime). Quinolone susceptibility was slightly increased, with a 2-fold reduction in MICs to levofloxacin and ciprofloxacin [84].

In addition, in order to study quinolone resistance, another CRISPR-Cas-based gene edition strategy was used. Due to the roll of *gyrA* as an essential gene involved in DNA uncoiling [88], its knocking down would compromise cell’s viability. Because of this, *gyrA* gene from the *P. aeruginosa* PA154197 strain was substituted by *gyrA* gene from the PAO1 strain, which exhibits MICs to both ciprofloxacin and levofloxacin of 0.25 µg/mL. The same approach was used with regulatory genes *mexR* and *mexT*, reverting mutations found in the PA154197 strain, which are absent in the quinolone-susceptible PAO1 strains. Results showed that a combination of the three gene substitutions (*gyrA^PAO1^*, *mexR^PAO1^* and *mexT^PAO1^*
strains) through the CRISPR-harboring plasmid pAY5235 yielded a 128-fold reduction in both ciprofloxacin and levofloxacin MICs. This was much higher than the *gyr^APAO1^* substitution alone, proving a synergistic effect of the three genes in quinolone resistance in the PA154197 strains [84].

Another approach when facing MDR infections is antivirulence therapy, which focuses on inhibiting the mechanisms through which bacteria communicate with each other (quorum sensing, QS), synthesize biofilm and toxins or arrange their functional membrane microdomains, rather than directly addressing antibiotic resistance [89]. Following this idea, in 2018, Chen and colleagues designed a dual-plasmid strategy (pCasPA/pACRISPR) which combined CRISPR-Cas technology with the λ-Red recombination system and a donor DNA template to edit different virulence regulatory genes [90]. These were the acyl-homoserine lactone receptor encoding genes *rhlR* and *lasR* (which are components of the QS systems in *P. aeruginosa*), the *nalD* gene (which encodes a repressor of the MexAB-OprM efflux pump), the pigment and QS regulators *rsaL* and *algR* (affecting, among others, pyoverdine and pyocyanin production) and the rhamnolipid synthesis regulator *rhlB* (involved in motility and biofilm disruption) [91,92,93,94,95,96,97].

Furthermore, given the toxicity of the heterologous Cas protein and the large size of the transforming plasmids, these authors developed an additional gene edition tool to increase transformation efficacy and broaden its applicability to other species within the *Pseudomonas* genus. This was achieved by engineering the pnCasPA-BEC plasmid, encoding for a Cas protein fused with a murine cytidine deaminase and several sgRNAs, which direct the Cas protein to the target sequence. As a result, *rhlR* and *rhlB* genes were successfully edited with an efficiency of 11/12 in both of them for the *P. aeruginosa* PAO1 strain and 14/15 and 17/17, respectively, for the *P. aeruginosa* PAK strain [90].

Finally, in 2020, Xiang and colleagues reduced the expression of the *prtR* regulatory gene in the *P. aeruginosa* PAO1 and PAK strains by transformation with the pHERD20T-dCas9-prtR plasmid [98]. This was achieved by engineering an inducible vector encoding for a dCas protein and the corresponding sgRNA and directed by an arabinose dependent promoter. As a result, transcriptome analysis in the *P. aeruginosa* PAO1 strain showed that in those strains in which *prtR* expression was inhibited, 902 genes were downregulated and 587 upregulated. These included the downregulation of genes related to alginate biosynthesis, iron acquisition, proteases and rhamnolipid synthesis, and the upregulation of genes associated with pyocin synthesis and other virulence factors such as the type 6 secretion system. This is consistent with the known targets of the *prtR* repressor [99].

### 3.6. Enterobacter *spp*.

*Enterobacter* spp. is a genus of Gram-negative bacilli which belongs to the normal human microbiota. However, this bacterium can act as an opportunistic pathogen in a variety of infections, especially those of nosocomial origin (sepsis, urinary tract infections, pneumonia or postsurgical peritonitis) [100]. This genus is characterized by the presence of a chromosomal AmpC β-lactamase, which depending on the degree of de-repression, can confer resistance to third generation cephalosporins, penicillins and their combinations with inhibitors and even ertapenem. In addition, these bacteria can acquire extended spectrum β-lactamases and carbapenemases through mobile genetic elements, considerably reducing the available therapeutic options.

In 2020, Hao and colleagues electroporated the engineered plasmid pCasCure (previously detailed for *K. pneumoniae*) to resensitize carbapenem-resistant *Enterobacter* species to these antibiotic family [66]. By targeting their *bla_KPC-3_*-harboring plasmids, the *E. hormaechei* 34978 and the *E. xiangfangensis* 34399 strains were cured with efficiencies up to 95.8 ± 2.1% and 95.1 ± 2.4%, respectively.

In the same year, Tagliaferri and colleagues targeted the plasmid-borne *bla_TEM-1_* gene in the *E. hormaechei* 4962 clinical isolate [101]. This plasmid, coding for the Cas9 protein and a specific sgRNA targeting the *bla_TEM-1_* gene, was successfully electroporated in the *E. coli* 189A clinical isolate, as seen by qPCR and phenotypic testing, with a re-sensitization to ampicillin, cefazolin, cefuroxime, ceftriaxone and cefotaxime. However, plasmid curing efficiencies in the *E. hormaechei* 4962 strain were considerably lower, with plasmid maintenance despite a substantial reduction in copies per cell. Furthermore, the concomitant presence of AmpC, CTX-M-9 and OXA-9 β-Lactamases within the *Enterobacter* species hampered phenotypic verification of the curing of plasmids.

## 4. Discussion

To date, only a few studies have been performed to analyze the ability of the CRISPR-Cas technology to modify genes related to antibiotic resistance or virulence factors in the ESKAPE group. These studies are mainly focused on in vitro experiments, and the goal of the experiment is often to edit bacterial genes independently of their function, just for the sake of the edition, rather than addressing AMR. In other experiments, the goal was to analyze the relationship between a specific mutation or gene with the acquisition of resistance to a particular antibiotic. The fact that CRISPR was not being seen by most researchers as a potential treatment but more as a genetic study tool explains why in some experiments MICs were not performed after CRISPR-directed gene edition and re-sensitization could not be studied.

Furthermore, some pathogens of the ESKAPE group such as *S. aureus* or *K. pneumoniae* have been studied more profoundly than others such as *A. baumannii*, *P. aeruginosa* or *Enterobacter* sp. (Table 1). It should also be noted that a wide range of studies focusing on *E. coli* [102,103,104,105,106] or other species of *Pseudomonas* [107,108] have not been included in this review as they are not members of the ESKAPE group. In other cases, studies were not analyzed because, although the studied bacteria belonged to the ESKAPE group, the edited genes were related to metabolic pathways rather than antibiotic resistance. Regarding *E. faecalis*, the studies performed by Dr. Palmer’s group were included in this review for their transcendence and extent, despite the fact that this species is not strictly included into the ESKAPE group [32,46,48,109].

One of the main reasons why CRISPR-Cas is still not considered as a potential antimicrobial treatment is the delivery issue. While in vitro plasmid electroporation is the method of choice to introduce the CRISPR-Cas system into the bacterial cells in the vast majority of the studies we present in this review, that would not always be possible to perform in vivo. In those cases, some other strategies are to be considered, such as phage-delivery and phagemids [110], conjugative plasmids [105] or polymeric nanoparticles [56] (Figure 5).

In the in vivo experiments we reviewed, conjugative plasmids were used for *E. faecalis* [48] and phage-delivery for *S. aureus* [25,53]. The obtained efficacies were sensibly lower than in the in vitro experiments, which is explained by the authors by the complexity of the environment in an in vivo model, with external factors affecting plasmid conjugation and the limited amount of phage which could be delivered into the infection site, respectively. Moreover, Rodrigues and colleagues proposed a novel strategy to treat patients colonized by MDR *E. faecalis* strains [48]. After editing these strains in vitro, researchers found them unable to regain resistance to erythromycin, opening the possibilities to probiotic treatments with CRISPR-Cas edited strains to gradually modify the patient’s microbiome. This approach resembles the one proposed by Ruotsalainen and colleagues [105], who designed “midbiotics” (plasmid-probiotics) targeting ESBL encoding genes. These authors highlighted the advantages of conjugative plasmids over phages, such as a broader host range and protection against the bacterium’s own restriction-modification system.

Another issue of concern when applying the CRISPR-Cas technology into the therapeutics field is the possible side effects of potential off-target modifications in the host’s genome, despite the specificity given by the PAM motive. In the first place, analyzing the host’s genome for potential similarities with the designed sgRNA should be an important anticipation step. On the other hand, using bacteriophages or phagemids as delivery constructs may serve to narrow the CRISPR-Cas system’s specificity thanks to phage tropism, thus avoiding its entry into the host’s cells. This could also be achieved by conjugative plasmids, which require both a donor and a recipient bacterium to be mobilized. In addition, immunity against the CRISPR-Cas system has also been seen as a potential risk. In their studies, Simhadri and colleagues analyzed 200 human serum samples and discovered the presence of antiCas9 antibodies, 10% against Cas9 proteins from *S. aureus* and 2.5% of *S. pyogenes* origin [111]. This could be a greater problem than delivery itself, not only due to the potential loss of efficacy upon Cas9 opsonization but also because of the immune response which could be triggered with the treatment. To answer these questions, further in vivo studies focusing on the safety of CRISPR-Cas antimicrobials and their interactions with the host’s immune system should be made.

To conclude, further studies should be performed to deepen the promising applications of CRISPR-Cas as an antimicrobial treatment, specially focusing in the in vivo experiments. The ability of CRISPR-Cas to target single bacterial clones, leaving the rest of the microbiome unaltered, or even to resensitize MDR bacteria without affecting their viability, contrasts with the collateral damage caused to the patient by broad-spectrum antibiotics. With further in vivo studies focusing on CRISPR’s efficacy in complex environments such as the gastrointestinal tract, potential off-target mutations, or the host’s immune response, CRISPR-Cas antimicrobials could be proven to be effective, pathogen-specific and secure.

## Figures and Tables

**Figure 1 antibiotics-10-00756-f001:**
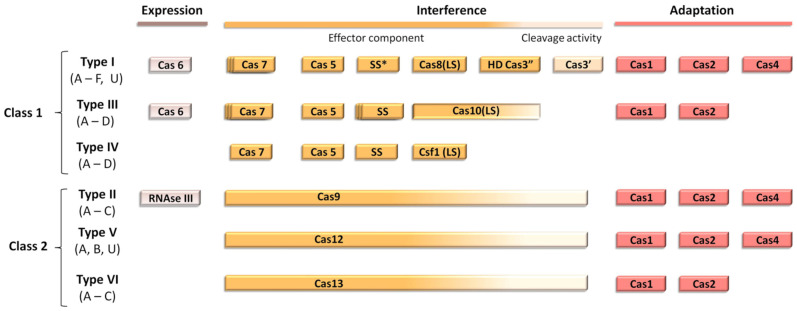
Modular organization of the different classes of CRISPR-Cas systems. Scheme adapted from Ishino et al. SS* indicates that the putative small subunit (SS) might be fused to the large subunit in several type I subtypes [13].

**Figure 2 antibiotics-10-00756-f002:**
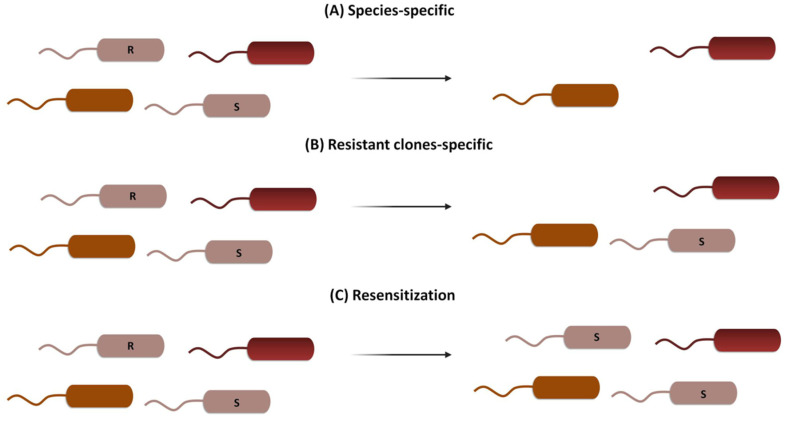
Three different CRISPR-Cas strategies to combat AMR. (**A**) Species-specific targeting kills both susceptible and resistant clones of the same species, leaving the rest of the microbial population unaltered. (**B**) Resistant clones-specific targeting kills only bacteria harboring genes for AMR, leaving susceptible clones and the rest of the population unaffected. (**C**) Resensitization turns resistant clones into susceptible ones by specifically targeting resistance genes, without an effect on the rest of the microbial population.

**Figure 3 antibiotics-10-00756-f003:**
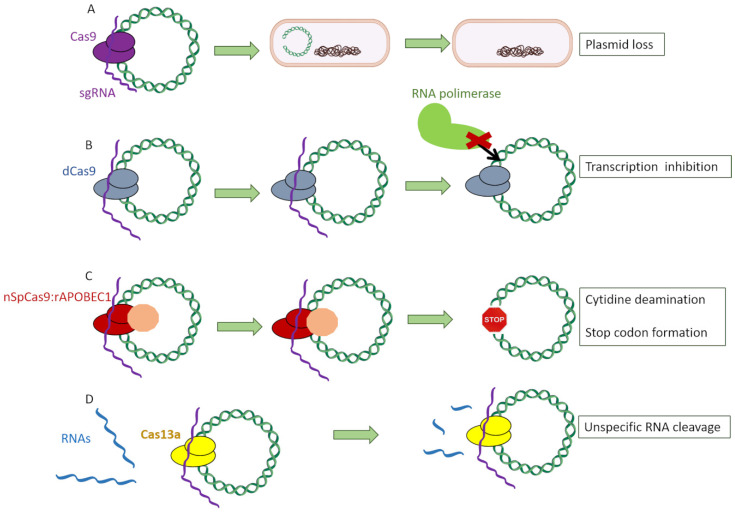
Different Cas proteins used to target antimicrobial genes: (**A**) Cas9, which specifically recognizes its target and induces a double-strand break. (**B**) dCas9, a defective Cas9 protein lacking the double-strand nuclease activity which specifically recognizes its target and stays attached to that region, hampering the binding of the RNA polymerase and thus the formation of the transcription preinitiation complex. (**C**) nSpCas9:rAPOBEC1, a Cas9 protein without nuclease activity fused to a deaminase, which causes the conversion of cytidine bases into thymine ones, thus creating a stop codon. (**D**) Cas13a protein, an RNA-specific endonuclease which indiscriminately cleaves RNA fragments upon activation by the recognition of its specific DNA sequence.

**Figure 4 antibiotics-10-00756-f004:**
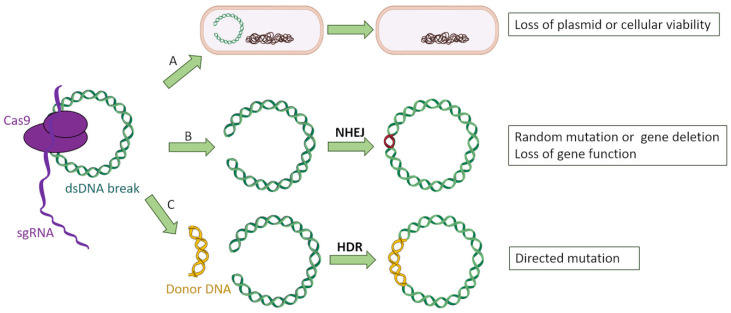
Three possible events after CRISPR-Cas9 targeting of bacterial genes: (**A**) The double-strand break affecting a plasmid leads to its loss, whereas a double break into the bacterial chromosome is lethal for the microorganism. (**B**) The bacterium attempts to fix the double-strand break by nonhomologous end joining (NHEJ), introducing random mutations into the targeted gene which causes a loss of function. (**C**) The bacterium uses a donor DNA fragment designed with the desired mutations to repair the double-strand break, incorporating those mutations.

**Figure 5 antibiotics-10-00756-f005:**
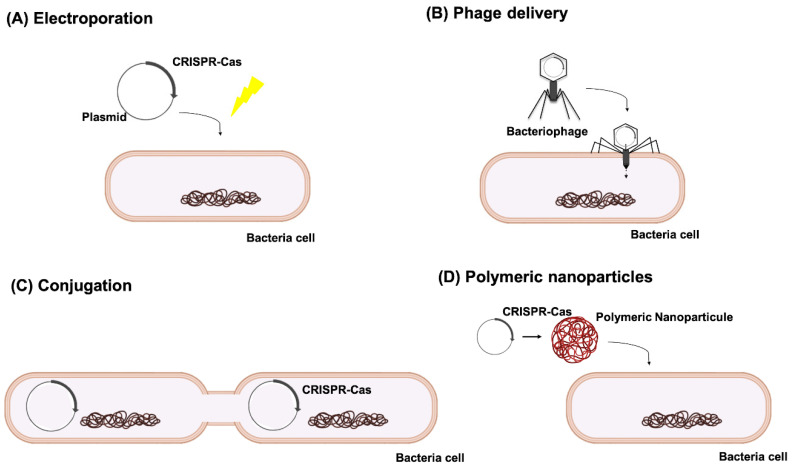
Different means of CRISPR-Cas delivery into the target cells: (**A**) plasmid electroporation, (**B**) phage delivery, (**C**) conjugation and (**D**) polymeric nanoparticles.

**Table 1 antibiotics-10-00756-t001:** Summary of the CRISPR-Cas strategies used in the treatment and study of the molecular mechanisms of AMR in bacteria belonging to the ESKAPE group.

ESKAPE Pathogen	Strain	CRISPR Strategy	Construct	Targeted Gene	Gene Function	Antibiotic/Virulence Factor Affected	Reference
*Enterococcus* spp.	*E. faecium* E745	HDR + intrinsic high recombination rates	Dual: pVLP3004 (Cas9 + tracRNA) and pVDM-*xmsrC* (crRNA + donor DNA template)	*msrC*	ABC transporter family efflux pump	Macrolides	[44]
	*E. faecalis* T11	Orphan CRISPR2 reactivation	PRP	pCF10	PRP	Antibiotic-resistance genes	[46]
	*E. faecalis* V583						[32]
	*E. faecalis* CK135 & OG1RF(∆*EfaRF1*) (Donor strains)	PRP conjugation + plasmid DSB	pKH88[sp-*tetM*] (Cas9 + tracRNA + crRNA)	*tetM*	Ribosomal protection protein	Tetracicline	[48]
	*E. faecalis* OG1SSp & V583 (Recipient strains)		pKH88[sp-*ermB*] (Cas9 + tracRNA + crRNA)	*ermB*	Ribosome methylation	Macrolides	
*S. aureus*	*S. aureus* USA300φ *S. aureus* RNφ	Chromosomal DSB	Phagemid pDB121	*mecA*	PBP2a	β-Lactams Cell’s integrity	[23]
	*S. aureus* AH1	Chromosomal DSB	pLI-158 pLI-252	*mecA*	PBP2a	β-Lactams Cell’s integrity	[51]
	*S. aureus* RN4220	HDR	pLQ-KI-ermR (Cas9 + sgRNA + donor DNA)	*ermR* *mecA*	23S rRNA methyl-transferase PBP2a	Erythromycin β-Lactams	[52]
	*S. aureus* CTH96	Chromosomal DSB	Phage φSaBov-Cas9-nuc (Cas9 + tracrRNA + crRNA)	*Nuc*	Thermostable nuclease	Cell’s integrity	[25]
	*S. aureus* 6538-GFP						[53]
	*S. aureus* ATCC 6538	Transcription inhibition	pLI50 (dCas9 + sgRNA)	*tarO* *tarG* *tarH*	Teichoic acid synthesis	Lysostaphin	[26]
	*S. aureus* ATCC 29213	Recombination and CRISPR counterselection	pCas9counter (Cas9 + sgRNA)	*rpoB*	RNA Polymerase	Rifampin	[57]
	*S. aureus* CCARM 3798, 3803 and 3877	Cationic polymer delivery + chromosomal DSB	SpCas9-bPEI (Cas9 + sgRNA + bPEI)	*mecA*	PBP2a	β-Lactams Cell’s integrity	[56]
	*S. aureus* USA300, USA300-∆*mecA* and RN4220	Phage capsid + indiscriminate ssRNA cleavage by Cas13a	pKLC-SP_*mecA* (Cas13a)	*mecA*	PBP2a	Bacterial transcription	[27]
*K. pneumoniae*	*K. pneumoniae* 5573	HDR + λ Red recombination	Dual: pCasKP (Cas9 + λ Red) pSGKP (sgRNA) + donor ssDNA	*fosA*	Glutathione transferase	Fosfomycin	[24]
		Cytidine deamination and stop codon formation	pBECKP (nSpCas9 + sgRNA)				
	*K. pneumoniae* KPCRE23	HDR + λ Red recombination	Dual: pCasKP (Cas9 + λ Red) pSGKP (sgRNA) + donor ssDNA	*bla* _SHV_	ESBL Carbapenemase	β-Lactams	
				*bla* _CTX-M-65_			
				*bla* _KPC-2_			
		Cytidine deamination and stop codon formation	pBECKP (nSpCas9 + sgRNA)	*bla* _KPC-2_			
	*K. pneumoniae* Y4	HDR + λ Red recombination	Dual: pCasKP (Cas9 + λ Red) pSGKP (sgRNA) + donor ssDNA	*mgrB*	LPS modification regulator	Colistin	[67]
	*K. pneumoniae* Y17	Cytidine deamination and stop codon formation	pBECKP (nSpCas9 + sgRNA)	*tetA*	Tetracycline efflux MFS transporter	Tetracycline	
		HDR + λ Red recombination	Dual: pCasKP (Cas9 + λ Red) pSGKP (sgRNA) + donor ssDNA	*ramR*	Efflux system regulator	Tigecycline	
	*K. pneumoniae* 13001	Plasmid DSB	pCasCure (Cas9 + sgRNA)	*bla* _KPC-2_	Carbapenemase	β-Lactams	[66]
	*K. pneumoniae* Kp97_58						
	*K. pneumoniae* 5193			*bla* _OXA-48-like_			
	*K. pneumoniae* 492110			*bla* _OXA-48_			
*A. baumannii*	*A. baumannii* XH386	Cytidine deamination and stop codon formation	pBECAb-apr	*bla* _OXA-23_	β-Lactamase	β-Lactams	[81]
				*bla* _TEM-1D_			
				*bla* _ADC-25_			
*P. aeruginosa*	*Pseudomonas aeruginosa* PA154197	Hampering native CRISPR-Cas I-F system + HDR	pAY5233 (sgRNA) + donor DNA	*mexB*	MexAB-OprM efflux pump component	β-Lactams Quinolones	[84]
				*mexF*			
				*mexH*			
				*gyrA*	Topoisome rase		
				*mexR*	MexAB-OprM efflux pump transcription regulation		
				*mexT*			
	*Pseudomonas aeruginosa* PAO1 *Pseudomonas aeruginosa* PAK	HDR + λ Red recombination	Dual: pCasPA (λ Red + Cas9) + pACRISPR (sgRNA + donor DNA)	*rhlR*	Acyl-homoserine Lactone receptor	QS	[90]
				*nalD*	Efflux pump repressor	Drug efflux pump	
				*lasR*	Acyl-homoserine lactone receptor	QS	
				*rsaL*	QS regulation Pyocyanin synthesisregulation	QS Pigment synthesis	
				*algR*	*rhlR* repressor	QS Pigment synthesis	
				*rhlB*	Rhamnolipid synthesis	Motility Biofilm disruption	
	*Pseudomonas aeruginosa* PAO1 *Pseudomonas aeruginosa* PAK	Cytidine deamination and stop codon formation	pnCasPA-BEC (SpCas9D10A + sgRNA)	*rhlR*	Acyl-homoserine lactone receptor	QS	
				*rhlB*	Rhamnolipid synthesis	Motility Biofilm disruption	
		Transcription inhibition	pHERD20T-dCas9-prtR (sgRNA + dCas9)	*prtR*	Pyocin synthesis repression	Pigment synthesis	[98]
*Enterobacter* spp.	*E. hormaechei* 34978 *E. xiangfangensis* 34399	Plasmid DSB	pCasCure (sgRNA + Cas9)	*bla* _KPC-3_	KPC-3 carbapenemase	Carbapenems	[66]
	*E. hormaechei* 4962	Plasmid DSB	pSB1C3 (sgRNA + Cas9)	*bla* _TEM-1_	TEM-1 betalactamase	β-Lactams	[101]

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
