# Peer review of "CRISPR-Cas, a Revolution in the Treatment and Study of ESKAPE Infections: Pre-Clinical Studies"

_antibiotics, 2021, doi:10.3390/antibiotics10070756_

Round 1

Reviewer 1 Report

The present review discusses the possibility of using CRISPR-Cas as tool for the fight of AMR, particularly for bacteria from the ESKAPE group. Strategies for the treatment and study of molecular mechanisms of AMR based on CRISPR-Cas are discussed and presented.

This review is timely and well performed and the schemes really help understanding the concept.

The major concern of this reviewer is related to the fact that, as it is written, AMR seems to concentrate only in bacteria, which is far from the truth. Over the years, the AMR for microorganisms other than bacteria - such as fungi - has been vastly increasing. Although this is not the focus of this work, it should at least be quickly addressed in the Introduction or the Conclusion.

Comments:

- Correct: “spp” in the Abstract, Line 251, 713 – remove italic form and add full stop;

- Please indicate how was the search and published works selection performed;

- In the section “ CRISPR-Cas: A new concept of antimicrobials”, this reviewer thinks that the MS could focus one small paragraph on the antimicrobial resistance problem (e.g.: doi.org/10.1099/acmi.ac2019.po0082; doi.org/10.1038/s41467-020-16731-6) and not only on section 3. Plus, the AMR should be extended to fungi., not only bacteria. In fact, the fungal AMR is growing as fast as bacteria and also needs to be addressed (even if it not the main focus of this MS, it needs to be, at least, mentioned);

- Line 285: Enterococcus faecalis – change to E. faecalis;

- Line 758: remove italic form on “sp”;

- These works should be considered in section 3 or in the discussion: doi.org/10.1016/j.jgar.2020.07.026; doi: 10.1038/s41559-020-1170-1;

- Line 251 - “Enterococcus spp”: remove italics on “spp” and add “.” (spp.);

- The table is missing a legend.

Author Response

Dear Ms. Irene Zhao, editors and referees of the Antibiotics journal,

Here we present the changes in our manuscript following the indications given by the reviewers.

Reviewer 1:

The present review discusses the possibility of using CRISPR-Cas as tool for the fight of AMR, particularly for bacteria from the ESKAPE group. Strategies for the treatment and study of molecular mechanisms of AMR based on CRISPR-Cas are discussed and presented.

This review is timely and well performed and the schemes really help understanding the concept.

The major concern of this reviewer is related to the fact that, as it is written, AMR seems to concentrate only in bacteria, which is far from the truth. Over the years, the AMR for microorganisms other than bacteria - such as fungi - has been vastly increasing. Although this is not the focus of this work, it should at least be quickly addressed in the Introduction or the Conclusion.

Comments:

- Correct: “spp” in the Abstract, Line 251, 713 – remove italic form and add full stop; corrected.

- Please indicate how was the search and published works selection performed; a small paragraph regarding the dates and keywords of the search has been included in section 3 (lines 258-263).

- In the section “ CRISPR-Cas: A new concept of antimicrobials”, this reviewer thinks that the MS could focus one small paragraph on the antimicrobial resistance problem (e.g.: doi.org/10.1099/acmi.ac2019.po0082; doi.org/10.1038/s41467-020-16731-6) and not only on section 3. Plus, the AMR should be extended to fungi., not only bacteria. In fact, the fungal AMR is growing as fast as bacteria and also needs to be addressed (even if it not the main focus of this MS, it needs to be, at least, mentioned); In order not to reduce AMR to bacteria solely, in the introduction the words "bacteria", "antibiotic" and "bacterial resistance" have been substituted for "microorganisms", "antimicrobials" and "AMR" respectively. An explicit mention to fungi has been made as well (line 115). The suggested publications have been also included. Finally, with the modifications applied to the first paragraph in section 2 (113-123) we believe that the AMR problem is already covered.

- Line 285: Enterococcus faecalis – change to E. faecalis; corrected.

- Line 758: remove italic form on “sp”; corrected.

-These works should be considered in section 3 or in the discussion: doi.org/10.1016/j.jgar.2020.07.026; doi: 10.1038/s41559-020-1170-1; papers have been added.

- Line 251 - “Enterococcus spp”: remove italics on “spp” and add “.” (spp.); corrected.

- The table is missing a legend. The legend was misplaced. It is now properly located above the table.

Reviewer 2:

The review presented is comprehensive and summarizes well the current state of the knowledge and experience in the field. While most of it balances well the information in the text with the figures provided, I find Figure 2 to be taking up too much space without significant contribution to the concepts presented in the text. My advice would be to completely eliminate it as the text itself is clear enough. Even if it might appear redundant, we would like to maintain Figure 2 since, in our opinion, it summarizes in a very visual manner one of the pillars of this review, the three general CRISPR-Cas approaches to AMR.

Form purely educational point of view adding another figure in Discussion section depicting main delivery options - phage-delivery and phagemids, conjugative plasmids or polymeric nanoparticles - would be very positive for the overall informativeness and quality of the review. As suggested, we have now added to the discussion a figure representing the four different ways to transport the CRISPR-Cas constructs into the target cells.

Besides, we have corrected the following typos:

            -Addition of "González" in the citation box on the right part of the cover, being the complete last name "González de Aledo".

            -Substitution of "e-mail@e-mail.com" for manugo04@gmail.com (M.G.A.) in the first affiliation.

            -Substitution of antimicrobial resistance for its acronym "AMR" in the Figure 2 legend (lines 149 and 151) and lines 252, 255 and 771.

            -Minor modification in Figure 3, which was lacking a plasmid with a double-strand break in the "a)" section.

            -The author Manuel González de Aledo is lacking his ORCID (https://orcid.org/0000-0001-7222-4941 )

We would like to thank you all for your feedback and comments.

Manuel González de Aledo and María Tomás

Reviewer 2 Report

The review presented is comprehensive and summarizes well the current state of the knowledge and experience in the field. While most of it balances well the information in the text with the figures provided, I find Figure 2 to be taking up too much space without significant contribution to the concepts presented in the text. My advice would be to completely eliminate it as the text itself is clear enough.

Form purely educational point of view adding another figure in Discussion section depicting main delivery options - phage-delivery and phagemids, conjugative plasmids or polymeric nanoparticles - would be very positive for the overall informativeness and quality of the review.

Author Response

(The authors gave the same response as above.)
